# Cluster detection with random neighbourhood covering: Application to invasive Group A Streptococcal disease

**Massimo Cavallaro**[1,2,3]*, **Juliana Coelho**[3], **Derren Ready**[3,4], **Valerie Decraene**[3], **Theresa Lamagni**[3], **Noel D. McCarthy**[1,5,6], **Dan Todkill**[3,5], **Matt J. Keeling**[1,2]

**1** The Zeeman Institute for Systems Biology & Infectious Disease Epidemiology Research, University of Warwick, Coventry, United Kingdom, **2** School of Life Sciences and Mathematics Institute, University of Warwick, Coventry, United Kingdom, **3** UK Health Security Agency, United Kingdom, **4** Health Protection Research Unit in Behavioural Science and Evaluation at the University of Bristol, Bristol, United Kingdom, **5** Warwick Medical School, University of Warwick, Coventry, United Kingdom, **6** Institute of Population Health, School of Medicine, Trinity College Dublin, University of Dublin, 2 Dublin, Ireland

* m.cavallaro@warwick.ac.uk

## Abstract

The rapid detection of outbreaks is a key step in the effective control and containment of infectious diseases. In particular, the identification of cases which might be epidemiologically linked is crucial in directing outbreak-containment efforts and shaping the intervention of public health authorities. Often this requires the detection of clusters of cases whose numbers exceed those expected by a background of sporadic cases. Quantifying exceedances rapidly is particularly challenging when only few cases are typically reported in a precise location and time. To address such important public health concerns, we present a general method which can detect spatio-temporal deviations from a Poisson point process and estimate the odds of an isolate being part of a cluster. This method can be applied to diseases where detailed geographical information is available. In addition, we propose an approach to explicitly take account of delays in microbial typing. As a case study, we considered invasive group A Streptococcus infection events as recorded and typed by Public Health England from 2015 to 2020.

## Author summary

Rapidly detecting disease outbreaks and identifying epidemiologically linked cases is crucial for the organization and implementation of effective outbreak-control interventions. To help identify the outbreak cases from a background of sporadic events—and thus complement the investigations conducted by public health teams—we developed a numerical approach, which we referred to as random neighbourhood covering (RaNCover). By summarising the statistical properties of the neighbourhoods of all events, RaNCover achieves excellent predictive performances. We applied our approach to invasive group A Streptococcus (GAS) infection events. While asymptomatic GAS carriage is prevalent, when these bacteria invade parts of the body such as the blood, the muscles, or the lungs, they

**Data Availability Statement:** Analyses were carried out using R ([44], version 3.4.3). t-distributed stochastic neighbour embedding was performed using the package Rtsne [52] with

default parameters. All software (also including codes to generate synthetic data) is archived online at https://github.com/mcavallaro/outbreak-detection. Data cannot be shared publicly as confidential. Data are available from UKHSA and must abide by the Caldicott principles. Material transfer must be obtained through a Material Transfer Agreement. Interested readers wanting similar data may wish to contact DataAccess@ukhsa.gov.uk, as detailed at https://www.gov.uk/government/publications/accessing-ukhsa-protected-data/accessing-ukhsa-protected-data.

**Funding:** This work was supported by Health Data Research UK, which is funded by the UK Medical Research Council, EPSRC, Economic and Social Research Council, Department of Health and Social Care (England), Chief Scientist Office of the Scottish Government Health and Social Care Directorates, Health and Social Care Research and Development Division (Welsh Government), Public Health Agency (Northern Ireland), British Heart Foundation and the Wellcome Trust (MC, MJK, and NDM). MJK and NM are affiliated to the National Institute for Health Research Health Protection Research Units (NIHR HPRUs) in Gastrointestinal Infections and in Genomics and Enabling Data. MJK is funded by UK Research and Innovation through the JUNIPER modelling consortium (MR/V038613/1). The views expressed are those of the author(s) and not necessarily those of the NIHR, the Department of Health and Social Care or UKHSA. The funders had no role in study design, data collection and analysis, decision to publish, or preparation of the manuscript.

**Competing interests:** The authors have declared that no competing interests exist.

can cause fatal infections. Invasive GAS outbreaks can take place across wide geographic areas and are difficult to detect using standard public health investigations, which can benefit from our novel computational approach.

## Introduction

Disease surveillance is the rapid collection and analysis of data to identify public health priorities, develop targeted response, and monitor the effects of interventions [1]. The importance of surveillance in controlling the spread of diseases is highlighted by several recent challenges, ranging from the fast evolving COVID-19 pandemic [2], to hospital-acquired anti-microbial resistant infections [3], to the burden of chronic non-communicable diseases [4].

For decades, statistical principles and methods have been leveraged by public health officials to identify increases in diseases and raise early warnings where necessary. The effectiveness of these methods has been improved by the availability of large digital databases, which can store infection records and quickly be interrogated and processed [5]. Thanks to these advances, syndromic surveillance is performed daily in the UK based on common symptoms such as cough, diarrhoea, or fever through general practitioner consultations [6–8]. In addition, the surveillance of microbiological diagnosis of specific pathogenic organisms is of key public health importance. Most of the existing statistical models for surveillance are tailored to dense temporal data streams, consisting of tens or hundreds of health records per week or day [9–15], although methods based on the so-called scan statistics [16–18] have good performance in detecting geographical areas with small confined outbreaks [19–27].

*Streptococcus pyogenes*, also known as group A Streptococcus (GAS), is a Gram-positive human pathogen that can colonise the throat and skin of humans. Asymptomatic throat carriage rates in the general population are reported to be around 7.5% in high-income countries [28]. Transmission of GAS is generally person-to-person, involving large respiratory droplets and secretions, and direct or indirect contact with carriers through mucous membranes or non-intact skin. Complex host-pathogen interactions in GAS infections can cause a broad range of symptom severity–from mild or asymptomatic infections to death. The most severe disease occurs when these bacteria invade parts of the body where they are usually not found (such as the blood, the muscles, or the lungs) and cause invasive infections, which requires hospitalisation and can lead to death [29]. An important part of the epidemiological surveillance of invasive GAS (iGAS) infections is the typing of collected bacterial isolates. A widely used typing method for GAS is based on the sequence of the N-terminal part of the M-protein gene (*emm*), which is a highly polymorphic gene and allows us to distinguish strains of *S. pyogenes* which would otherwise look identical. So far, more than 200 *emm* types have been identified in Europe and North America, with few common types contributing to a high percentage of the total number of iGAS cases [30]. In the UK, iGAS infections are notifiable and isolated from iGAS cases are sent to the national reference laboratory for *emm* typing, enabling the identification of cases which share the same *emm* gene and which could denote an outbreak. Non-invasive GAS isolates are rarely typed outside of an outbreak investigation. iGAS disease is of particular concern due to the recent upsurge in the number of cases in some countries [31] and potential for spread amongst marginalized groups [32] or in community nursing teams [33]. In these latter two settings, outbreaks can take place across wide geographic areas and are therefore more difficult to detect using standard public health investigations.

To help identify clusters of iGAS which share the same *emm* type and are linked by geography and in time, we developed the random neighbourhood covering (RaNCover) approach.

As an effective and automated algorithm, it would be complementary to what local health protection or public health teams can offer in the initial stages of outbreak detection, and may be of particular use with outbreaks which cross regional or team borders that might otherwise be difficult to detect. The general problem RaNCover addresses is to classify a single recorded infection as being part of a cluster or being consistent with a baseline of sporadic cases. To this end, we consider a neighbourhood of the point location and time of the recorded case and draw a random collection of spatio-temporal cylinders to cover it. Based on the properties of this collection of covering cylinders it is possible to perform classification. RaNCover is specifically tailored to diseases where detailed spatio-temporal information is available and aggregating data in coarse-grained bins (which results in less detailed information) is not desirable. As proof of concept, we deployed our framework on *emm* typed iGAS infections recorded in England.

The manuscript is organised as follows. In section *Methods* we introduce the outbreak-detection principles, define the RaNCover approach, illustrate it on simulated data, introduce the *emm* typed iGAS data, and derive the endemic baseline for this disease. The latter is used to generate synthetic data and perform additional simulation experiments for validation. In the section *Results*, RaNCover is retrospectively and prospectively applied to the iGAS data and the detected clusters are discussed with the help of a dimensionality reduction technique, which allows us to conveniently present the results. In the same section, we tackle *emm* typing delay. We conclude with a *Discussion* section.

## Methods

### Ethics statement

The data used in this study were supplied from the UK Health Security Agency (UKHSA) under strict data protection protocols agreed between the University of Warwick and UKHSA. The ethics of the use of these data for these purposes was agreed by UKHSA.

### Null model and outbreak detection

A spatio-temporal disease pattern can often be decomposed into two components, endemic and epidemic—although such a distinction can be blurred [34]. The endemic component explains a baseline rate of cases with a persistent yet stable pattern, only reinforced by immigration [35]. The epidemic element can be thought of as realizations of processes with autoregressive or self-exciting behavior, which arise from a common source, e.g., environmental exposure, increased susceptibility in the population, or invasion of a new pathogen. In other words, the presence of epidemic events is correlated with and reinforces the occurrence of successive events [34].

We argue that the most important characteristic common to all epidemic data is that observations are not independent. Therefore, to state whether cases are observed in atypical frequency, we must compare our observations with those expected if the cases were uncorrelated in both space and time. If the recorded case observations are consistent with such an uncorrelated baseline, no warning should be raised. This suggests that the first step of an outbreak detection system is to develop a naïve model for the observed cases—a *null model*—which does not consider any spatial or temporal correlations. The second step is then to detect (statistically significant) deviations of the observed pattern from that produced by the null model.

We let $X(A)$ denote the random variable that describes the number of cases detected in a region of space and time ($A \subset \mathcal{S}$) under the null model ($X$) and define a threshold value $\theta_A$

such that

$$\mathrm{Prob}(X(A) > \theta_A) = \alpha. \qquad (1)$$

A basic exceedance criterion therefore consists of triggering a warning flag if the number of cases observed over $A$ exceeds a value $\theta_A$. Here, $\alpha$ is the false-positive detection rate, which is the probability of triggering a warning by chance when no outbreak has occurred, and the set $\mathcal{S}$ represents the entire extent of the spatio-temporal data. The approach described so far is not new [5,36] and can be regarded as the standard in statistical process control [37].

## The random neighbourhood covering approach

We assume that the null model is an inhomogeneous Poisson point process with intensity $\lambda(x, y, t)$, where $x$ and $y$ are spatial coordinates and $t$ is the time. We stress that the rate parameter $\lambda$ is not necessarily constant in space or time. Indeed, even in the absence of localized outbreaks, we may expect many episodes in highly populated regions and at certain times of the year, as many diseases are known to be influenced by seasonality while weekends or holidays can impact the daily recording practice [10,13] [38]. A Poisson point process across space and time ($\mathcal{S}$) is defined by the following two properties:

1. $X(A_1)$ and $X(A_2)$ are independent for disjoint subsets $A_1$ and $A_2$ of $\mathcal{S}$;

2. For any subset $A \subset \mathcal{S}$, $X(A)$ has Poisson distribution

$$\mathrm{Prob}(X(A) = n) = \Lambda(A)^n e^{-\Lambda(A)}/n!, \qquad (2)$$

$$\Lambda(A) = \int_A \mathrm{d}\xi \lambda(\xi),$$

where $\xi = (x,y,t)$ and $\Lambda(A)$ is referred to as the intensity measure [39]. Given the intensity function, it is straightforward to numerically obtain $\Lambda(A)$, find the $\alpha$th quantile $\theta$, and state whether the observed number $n$ exceeds this value in the subset $A$. Those subsets such that the observed event number exceeds $\theta$ are tagged as outbreak subsets. For simplicity, it is convenient to treat the time $t$ as an integer variable rounding it up to, e.g., days or weeks, thus $\mathcal{S}$ is $\mathbb{R}^2 \times \mathbb{N}$, its generic subset $A$ is the union $\cup_{i \in \mathbb{N}} A_i$, $A_i = \{(x,y,t) \in A | t = i\}$, and the intensity measure can be easily computed as sum of two-dimensional integrals, i.e., $\Lambda(A) = \sum_i \int_{A_i} \mathrm{d}x \mathrm{d}y \lambda(x, y, i)$.

The intensity function $\lambda$ represents the average trend of the endemic infections. We assume that it factorises into a spatial factor $\lambda_0$ and a temporal factor $\lambda_{\mathrm{time}}$, i.e., $\lambda(x,y,t) = \lambda_0(x,y) \lambda_{\mathrm{time}}(t)$, thus excluding any correlation between the space and time trends. We make the simplifying assumption that $\lambda_0(x,y)$ is proportional to the population density, reflecting the fact that the number of observed episodes in an area is likely to increase with its population size–although $\lambda_0(x,y)$ could be matched to historical infection data if these were plentiful. This population-based weighting is arguably the most important factor in the spatial distribution of the episodes, although, across geographical areas, other factors such as the climatic zone or disparities in access to health care could affect the baseline. The temporal factor encodes the disease seasonality, and other long-term trends, and is generally obtained by fitting a periodic function to available data.

The inequality (1) provides a basic exceedance criterion for flagging a subset $A$ (which encompasses a geographical region for a certain time interval) but is far from being ideal. Indeed, a warning is raised with probability $\alpha$ even if the data is actually generated by the baseline model $X(A)$. Further, the aberration can only be tested on the subset $A$ and outbreaks

diffused over a larger area are likely to be missed [40]. To address these issues we consider, for each reported case (of given space-time coordinates $\mathbf{x} \in \mathcal{S}$), $N$ overlapping subsets $A_i \subset \mathcal{S}$, i = 1,2,$\cdots$,$N$, each of which contains the case ($\mathbf{x} \in A_i$) and apply the criterion of (1) to each one of them in order to test whether there is aberration from the baseline Poisson point process (Fig 1). In the absence of any aberrations and in agreement with property 2, the number of detected cases in any subset $A_i$ has distribution (2). If its intensity measures $\Lambda(A_i)$ are the same for all the subsets, we expect just a fraction $\alpha$ of them to exceed the threshold $\theta_{A_i}$ (while in the presence of epidemic events the fraction is greater than $\alpha$). Throughout this manuscript, we stick to the standard false discovery rate $\alpha = 0.05$ for each single subset. It is worth noting that, by considering the state of multiple subsets, we increase robustness to random fluctuations and make highly unlikely to identify a cluster from random case data. In the null model, we indeed expect just 5% of independent subset to have a warning flag and even with just 20 subsets, the probability of generating a false flag is very small (probability of 19 out of 20 subsets having a flag is $3.6 \times 10^{-24}$).

For a detected case $\mathbf{x}$, we refer to the proportion $w(\mathbf{x})$ of flagged subsets $A \ni \mathbf{x}$ as its *warning score*. By taking this approach, we are able to go beyond the binary warning flag for an outbreak event, and hence present $w(\mathbf{x}) \in [0, 1]$ as a continuous warning score specific to a detected case, high values of $w$ corresponding to a high probability that an outbreak is occurring. This suggests raising a warning if $w(\mathbf{x})$ is significantly greater than a chosen threshold, where the statistical significance is tested by means of a binomial test for proportions. Wilson intervals [41] are used as 2.5%-97.5% confidence intervals (CIs) for the warning scores. As $N$ increases, these CIs tighten and the test significancy improves thus allowing practitioners to tune the parameter $N$ and reach their desired significance level.

It is convenient to focus on subsets placed over detected cases, thus testing for aberration only where necessary (Fig 1). This obviously implies that every subset always contains at least one point event and its null-model prediction is shifted to $1+X(A)$. In space-time, we opt to select cylinders of the same volume $\int_A d\xi$. In order to allow cylinders of the same volume to enclose outbreaks highly diffused in space or lasting long in time, we randomly generate cylinders of variable radii $\varrho$ and heights $h$, still subject to the constraint that their volume $\pi\varrho^2 h$ is

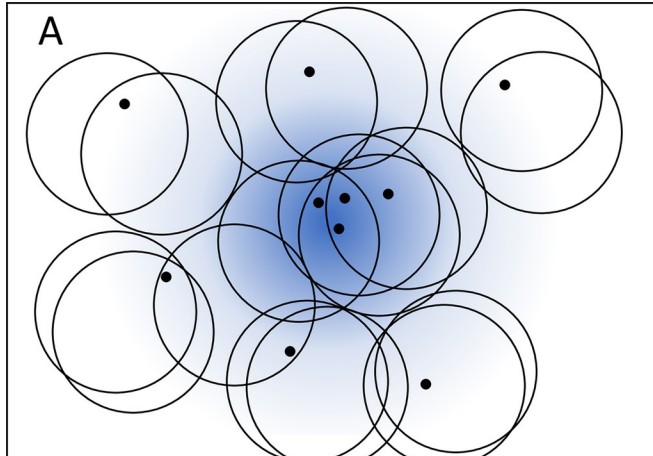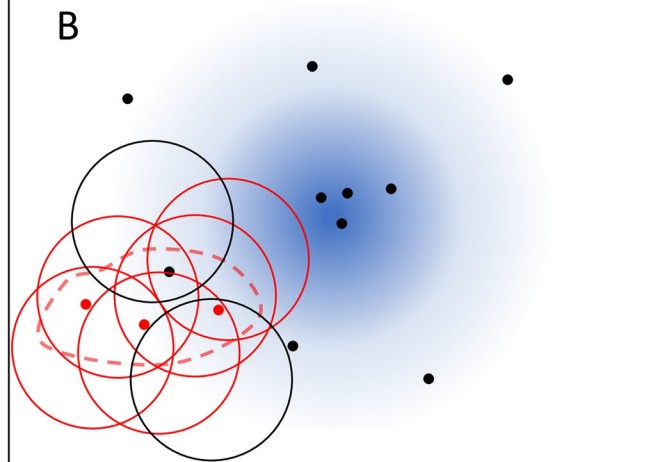

**Fig 1. Representation of random neighbourhood covering on a two-dimensional space. A** Black markers represent point events that are generated from an endemic baseline (blue shade). Circles are randomly drawn to cover the neighbourhoods of the events. The number of events in any circle is consistent with the endemic baseline prediction within statistical fluctuations. **B** Three point events (red markers) representing an unexpected outbreak are on the region delimited by the dashed line. Many circles (coloured in red) in this region contain significantly more points than the baseline prediction.

constant. The algorithm appears to be robust to the choice of this volume (see simulation experiments); its value is heuristically set to avoid excessively small volumes, expected to contain only one event, and excessively large volumes, which would contain more cases but might not represent well local information.

It is worth noting that a cylinder can accommodate many point events and nearby events are likely to be enclosed by the same cylinder. Consequently, the warning scores are correlated and their values reflect the neighbourhood structure of the detected cases, as demonstrated in the case study of section *Results*. Also, in simulation experiments performed for validation, clusters are detected as groups of cases with high warning scores. To summarise, for a single infection case **x**, RaNCover returns a warning score $w(\mathbf{x})$ that measures to what extent its presence alters the local count statistics from those of a Poisson point process of intensity $\lambda$. An appropriate baseline intensity needs to be estimated or given as an input.

We now illustrate the method in action by means of synthetic data and baseline. To this end, we assume the baseline intensity $\lambda = \lambda_0(x,y)\lambda_{\text{time}}(t)$ with

$$\lambda_0(x, y) = (C + De^{-E((x-0.5)^2 + (y-0.5)^2)}), \tag{3}$$

$$\lambda_{\text{time}}(t) = \left(\frac{A}{2}(\sin(t) + 1) + B\right) \tag{4}$$

over a unit square region with parameters $(A,B,C,D,E) = (7,0.9,0.5,20,80)$ where $x$ and $y$ are spatial coordinates, while $t$ is the time index ranging from 1 to 20. This describes a spatial distribution of cases peaked at the centre of the region, modulated by a periodic temporal trend. For each value of $t$ we simulate a spatial Poisson point process with intensity $\lambda$, which yields a random pattern of points representing endemic baseline episodes. An outbreak in a delimited polygonal region is simulated by randomly generating additional Poisson point events with intensity $50\,\lambda_{\text{time}}(t)$ for $t$ = 5,6,7,8,15,16,17,18 and zero elsewhere. Spatial Poisson processes are generated using the thinning method [42]. The resulting point pattern is illustrated in Fig 2A for the first 9 time steps. To identify the events corresponding to the simulated outbreak in this point pattern we apply RaNCover as follows. We draw $N$ = 10,000 random covering cylinders of equal volume, with radii $\rho \cong (0.10,0.12,0.14,0.17,0.23)$ and height $h = (1,2,3,4,5)$. The chosen volume is $\pi\rho^2 h \cong 0.17$, which corresponds to about 5 expected baseline events per cylinder. The centre coordinates are randomly sampled with replacement from the episode locations and times, and randomly shifted whilst ensuring that each cylinder contains its assigned episode location. We then flag all the cylinders according to the basic exceedance criterion with $\alpha$ = 0.05. The classification performed by the algorithm is summarised by the warning scores, computed using this set of cylinders. These warning scores indeed appear to capture the synthetic outbreak, as illustrated in Fig 2B.

## Performance metrics

Excellent performance is confirmed by the receiver operator characteristic (ROC) curve (panel A in S1 Fig) and the area under the curve (ROC-AUC 0.90, 95%CI 0.85–0.95 by means of bootstrap). The warning scores obtained upon increasing the volume of the cylinders by 40% achieve close performances (ROC-AUC 0.92, 95%CI 0.87–0.96) and are strongly correlated with the first replicate output (Pearson $\rho$ = 0.95, P = $1 \times 10^{-83}$), thus suggesting that the result is robust to the choice of the cylinder size (panels B and C in S1 Fig).

In a more realistic surveillance scenario, the warning scores are prospectively updated as soon as new cases are detected and included into the dataset, which we take to occur at every point in time. Observing the individual warning scores varying as new observations are

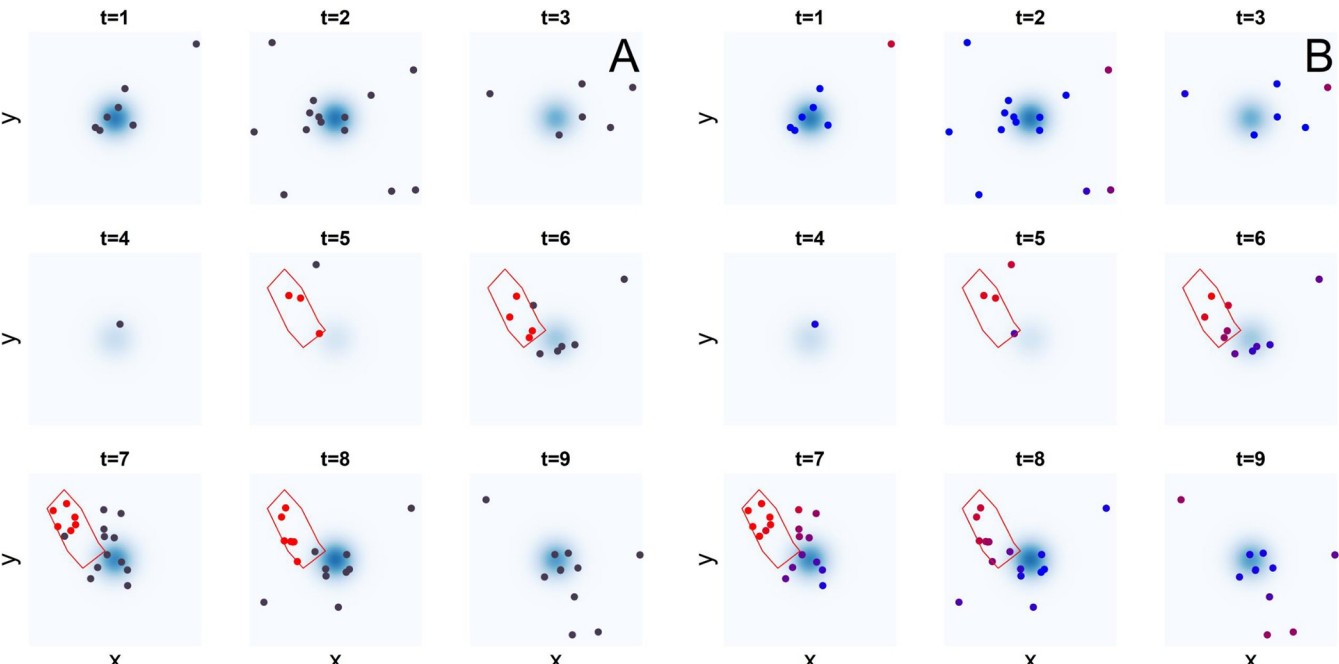

**Fig 2. Simulation experiment. A** Synthetic baseline point events (black markers) for $1 \leq t \leq 9$. Additional points representing an outbreak (red markers) are in the areas delimited in red. **B** Algorithm prediction. Episode markers are coloured by their warning scores, blue ($w = 0$) to red ($w = 1$). The algorithm assigns high warning scores to the events in the outbreak areas.

included allows us to assess timeliness. This measures the time elapsed between the case detection day and the day at which its warning score stabilises around its final value (corresponding to the retrospectively computed warning score). To simulate this, we iteratively apply the procedure detailed above to all the observations occurred until time $\tau$, $\tau = 1, 2, \ldots, 20$ and observe how the warning scores of cases detected previous to the update time $\tau$ vary with $\tau$, while new cases are included and assessed. Fig 3 shows this process in action, the three panels show the warning scores for outbreak and endemic cases (as defined by the Poisson intensity function) that are detected at times $t = 5$, $t = 6$, and $t = 7$. For this simulation, the values of the warning scores appear to converge after only one or two updates, which suggests that the method can reach excellent timeliness.

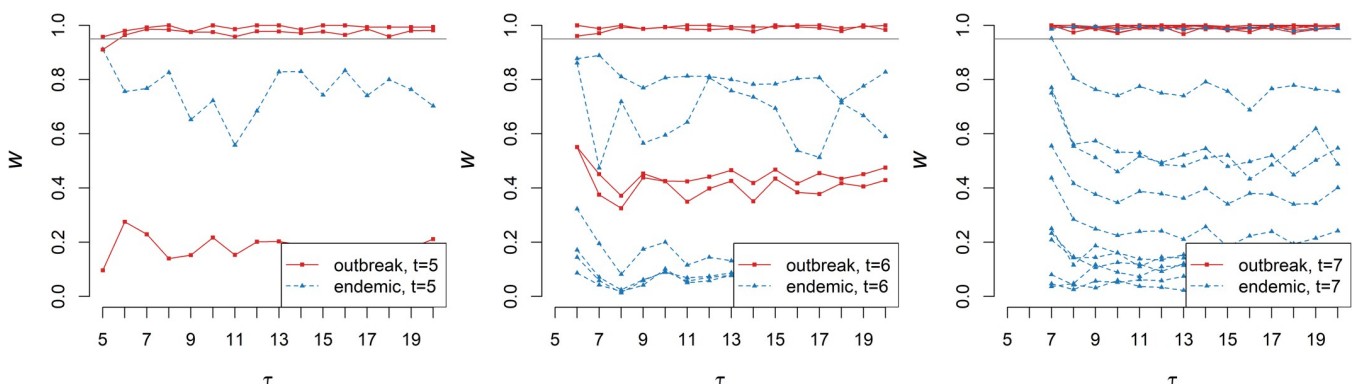

**Fig 3. Timeliness.** As the time $\tau$ progresses and new cases and cylinders are added, the warning-score values of cases previously detected (here at $t = 5, 6, 7$, see also Fig 2) are updated. In fact, these converge quickly and even the earliest warning values are informative.

### Defining iGAS background rate and reporting delay

Our study cohort consists of 10,820 iGAS records comprising 465 different *emm* subtypes, with the three most prevalent types (viz., 1.0, 89.0, and 12.0) accounting for more than 38% of all records, while other *emm* types occur one order of magnitude less frequently (Fig 4). A total of 235 different *emm* subtypes were recorded only once in the cohort.

We estimated the endemic baseline for iGAS as follows. As streptococcal isolates of different *emm* types are thought to belong to different endemic populations, they cannot form a single outbreak even when they occur in the same geographical location and at the same time [31]. Multiple emm type infections are very uncommon, and our iGAS database does not report presence of patients infected with different emm types. To encode such mutual independence, the baseline intensity $\lambda_M(x,y,t)$ specific to the *emm* type M is obtained by multiplying $\lambda(x,y,t)$ by the relative frequency $\lambda_m$ of M in the population, computed retrospectively until time $\tau$. In other words, the null-model intensity function is assumed to obey

$$\lambda_M(x, y, t) = \lambda_m \lambda_0(x, y) \lambda_{\text{time}}(t). \tag{5}$$

The temporal and spatial factors ($\lambda_{\text{time}}$ and $\lambda_0$, respectively) remain the same for all *emm* types. As the incidence of iGAS infection is subject to seasonality [29], $\lambda_{\text{time}}(t)$ is chosen to be periodic with a period $T$ of one year, viz.,

$$\lambda_{\text{time}}(t|\vartheta_\tau) = a + b\,t + (c + b\,t)\,\sin(2\pi d/T + 2\pi t/T), \tag{6}$$

where $\vartheta_\tau := (a,b,c,d)$ are parameters that must be retrospectively estimated from the data until time $\tau$. To determine these parameters, we use the principle of maximum likelihood estimation (MLE). We denote by $t_1, t_2, \ldots, t_n$ the ordered times of all cases detected until $\tau$, $0 < t_1 < t_2 < \cdots < t_n < \tau$, regardless of their *emm* types and geographical locations. The parameter $\tau$ identifies the time interval used to estimate the temporal trend of the baseline; for iGAS this has to be long enough to capture seasonality, with longer $\tau$ arguably producing better estimates. The likelihood function $L(\vartheta_\tau)$ under the Poisson null model is given by

$$\log L(\vartheta_\tau) = - \int_0^\tau \lambda_{\text{time}}(u|\vartheta_\tau)\mathrm{d}u + \sum_{i=0}^n \log\lambda_{\text{time}}(t_i|\vartheta_\tau).$$

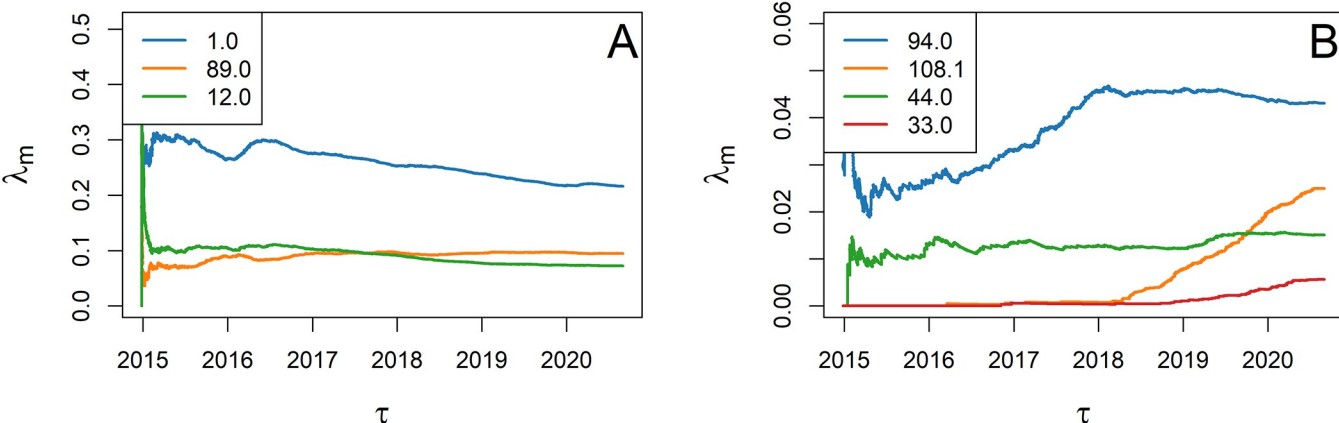

**Fig 4.** Relative frequency $\lambda_m$ of selected emm types in England computed retrospectively over the observation times $\tau$. **A** The fraction the total of three most common types (viz., 1.0, 89.0, and 12.0) is either overall declining or stable, yet they still account for more than 38% of all episodes at the end of the study. **B** Selected emm types show concerning relative growth (108.1 and 33.0) or stable pattern (44.0 and 94.0) over recent years.

The value of $\vartheta_\tau$ that maximises the likelihood was recursively obtained by maximising over $a,c,d$ using the Nelder-Mead method with $b$ held fixed and then finding the global maximum over $b$ with the other parameters held fixed.

The episodes in the database are labelled by patient postcode location. For each postcode, the estimated population size is obtained from the 2011 Census in England and Wales by the Office for National Statistics (ONS), which allows us to tabulate the values of the spatial baseline intensity $\lambda_0(x,y)$. The corresponding spatial coordinates are taken to be the postcode centroid locations, also retrieved from the ONS. This completes the definition of the endemic intensity function $\lambda(x,y,t)$.

The available iGAS records also include the week at which the original sample was taken, which, on average, occurred a week earlier than the laboratory *emm* typing. The time elapsed between the sample date and the *emm*-type report is referred to as the reporting delay. We generalise RaNCover to include untyped cases by introducing the baseline intensity $\lambda_U := \lambda_u(t;\tau)\lambda_{\text{time}}(t)\lambda_0(x,y)$ of untyped cases at each update time $\tau$, where $\lambda_u(t;\tau)$ is the average fraction of cases detected at time $t$ and *emm* typed after time $\tau$. For a generic spatio-temporal subset $A$, we refer to the number of typed observations, the typed baseline prediction, the number of untyped observations, and the untyped baseline prediction as $n_M(A)$, $X_M(A)$, $n_u(A)$, and $X_u(A)$, respectively. The next step is to define a strategy based on the standard exceedance criterion (1). We flag $A$ if $n_M(A)$ exceeds the threshold value $\theta_{M,A}$ or $n_u(A)$ exceeds the value $\theta_{u,A}$, i.e.,

$$(n_M(A) > \theta_{M,A}) \vee (n_u(A) > \theta_{u,A}),$$

with thresholds defined by $\text{Prob}(X_M(A) > \theta_{M,A}) = \alpha$ and $\text{Prob}(X_u(A) > \theta_{u,A}) = \alpha$, respectively. $X_M(A)$ and $X_u(A)$ are assumed to be Poisson point processes of intensities $(1-\lambda_u(t;\tau))\lambda_M(x,y,t)$ and $\lambda_U(x,y,t;\tau)$, respectively. The warning score of an untyped case $\mathbf{x}$ is the fraction $w(\mathbf{x})$ of subsets $A \ni \mathbf{x}$ flagged accordingly to this criterion.

## Results

### Simulation study

Benchmarking surveillance systems benefits from realistic simulations of health data [43,44]. As described in *Methods*, we assume that, in the absence of outbreaks, the detected cases are a realization of a generative Poisson point process with factorised intensity function $\lambda_0(x,y)$ $\lambda_{\text{time}}(t)$. Accordingly, we simulated the point process with intensity defined by Eqs (2) and (3) and parameters as estimated in section *Methods*, thus yielding a synthetic baseline of 4915 endemic cases (distributed over a period of 100 weeks in England) that should not raise new public-health concerns. The temporal intensity function of Eq (6) fitted to all data in the study period is illustrated in Fig 5. We simulated an outbreak by including additional (epidemic) point events localised in 19 selected postcode areas (arbitrarily chosen around the town of St. Albans, north of Greater London) during a chosen time interval (weeks 40 to 59). In order to imitate the initial spread of infections and their subsequent control, the total number of epidemic events per week is initially low, increases and peaks around week 50, and finally decreases until the number of epidemic cases drops to zero (Fig 6A). Aggregating the cases over their geographic locations clearly shows the temporal trend as illustrated in panel A in S2 Fig for the endemic cases and in panel B in S2 Fig for both endemic and epidemic cases. Based on these time series, it is virtually impossible to detect excess cases during the outbreak. Analogously, aggregating over the time yields the geographical distributions of cases, which mainly occur alongside the most populated area (panel C in S2 Fig) and show no appreciable changes if the outbreak in St. Albans is included (panel D in S2 Fig). Yet, whether a single case is part of

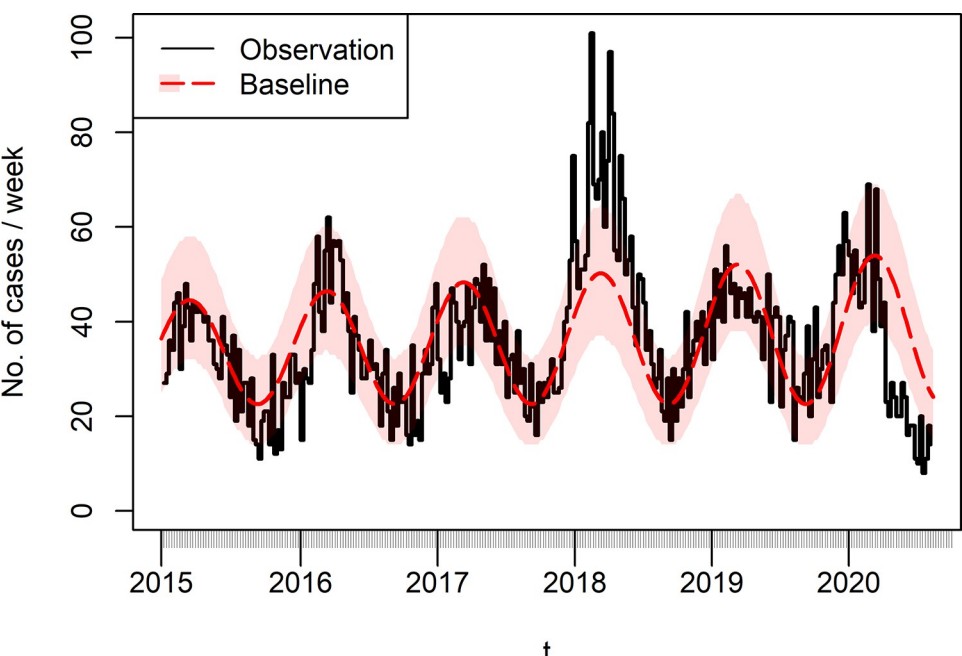

**Fig 5.** Weekly number of iGAS infection cases (black solid line) and fitted temporal baseline function $\lambda_{time}$ (dashed red line, shaded area comprises 95% confidence interval). 2018 was a year of heightened transmission across England, as reported in [56]. The overall fitted trend is increasing, despite the sharp drop in 2020.

the outbreak or not is clearly labelled in the synthetic dataset, thus making it ideal for evaluation purposes; this typically is not the case in real data.

The set of these simulated data was used to test whether RaNCover can detect the epidemic components in it. RaNCover for retrospective analysis returns a warning score $w(\mathbf{x})$ for each case $\mathbf{x}$. We set $\alpha = 0.05$, $N = 1,000,000$, and cylinder volumes $\cong 2.605\ km^2 \times week$, which overall yielded tight CIs (S3 Fig). In Fig 6B, each warning score is plotted vs the case detection time $t$. In this figure, the markers are coloured in blue if the case was simulated from the baseline and red if simulated from the outbreak. It is possible to appreciate that the baseline events have low warning scores within statistical fluctuations. The bulk of the outbreak events always had warning scores close to one and can be unambiguously discriminated from the baseline based on $w(\mathbf{x})$. Unsurprisingly, few cases at the tails of outbreak can be confused with background, but their warning scores reach relatively large values ($w(\mathbf{x}) > 0.8$). The ROC-AUC is 0.99 (95% CI 0.98–1.00), thus suggesting excellent performances. In Fig 6C, we show that the method can also identify the geographic location of the outbreak with precision. The result appears robust to the choice of the cylinder volume, as suggested by the fact that the warning scores calculated from a second replicate where cylinder volume was increased by 40% are strongly correlated with the first replicate (Fig 6D).

As a further test we sequentially calculated the warning scores prospectively, using only the data simulated until the week $\tau$, $\tau = 40,41,\cdots,59$. As time progresses and new cases are detected, the warning scores are updated, thus allowing us to observe how quickly the outbreak is detected. The warning-score traces for the epidemic cases are plotted in Fig 6E, where it is possible to appreciate that, for the episode that occurred at t = 40 (which is the first outbreak event and appears isolated at first, see Fig 6A), the warning score is initially very low and increases when new outbreak cases occur (until it converges to a higher stable value). Similarly, the other warning scores become stable after few steps.

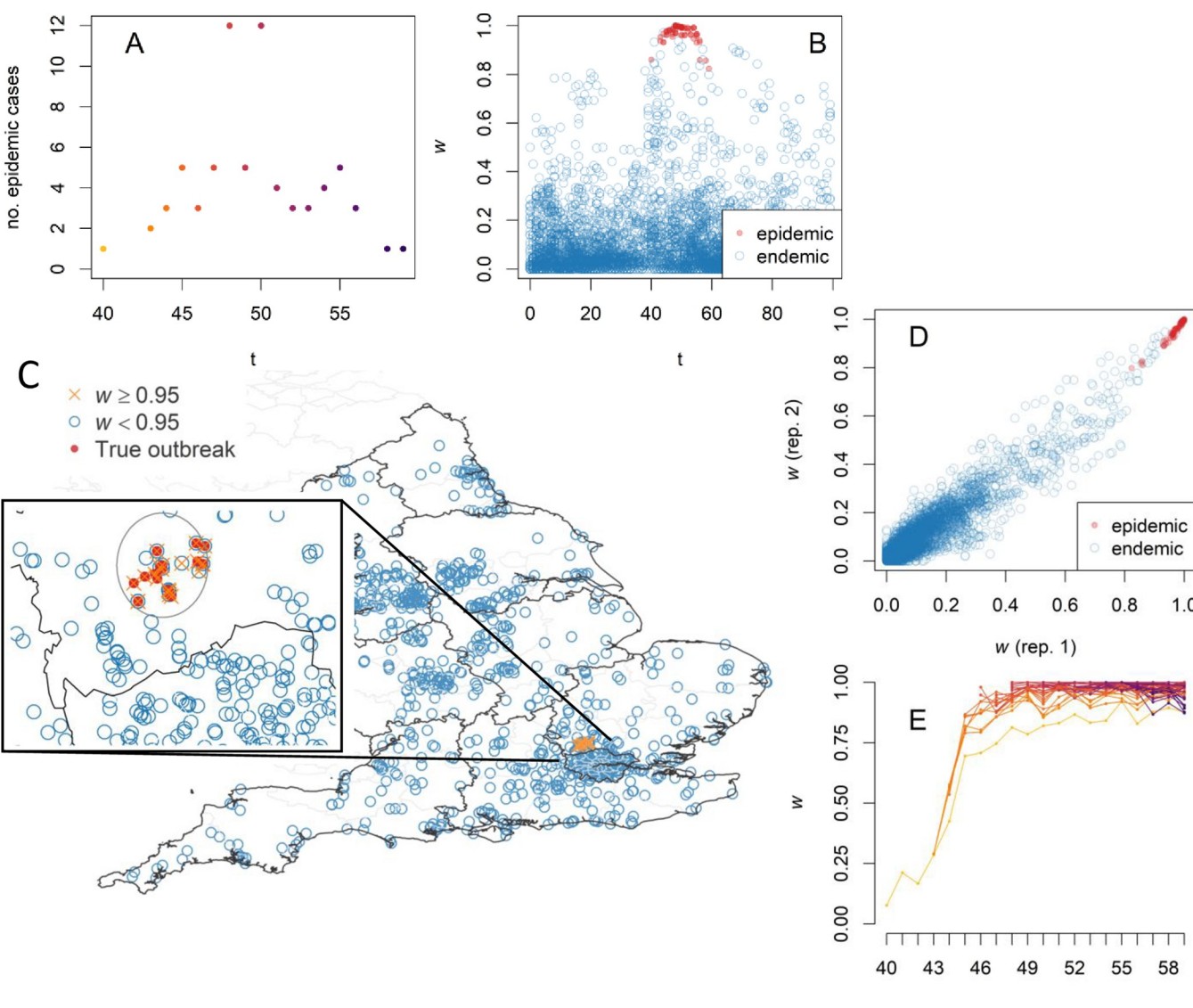

**Fig 6. Simulation experiment on realistic baseline. A** The total number of synthetic epidemic cases increases slowly from t = 40 until it reaches a peak at week t = 48, thus simulating the emergence of an outbreak. **B** The retrospectively-computed warning scores w(**x**) for the epidemic cases (red markers) are typically larger than those for the endemic cases (blue markers); plotting these vs time highlights the epidemic cluster. **C** Simulated outbreak points are close in both time and space, but it is hard to naively detect the anomaly (see also S2 and S4 Figs); localising the cases with w(**x**)>0.95 (orange markers) identifies the outbreak epicentre; in the inset, the true outbreak cases are marked with a red cross and the area detected by SaTScan is circled for comparison. Map created with Sf [55] using shapefiles from the GADM database (https://gadm.org/maps/GBR_1.html) and the Ordnance Survey Data Hub (https://osdatahub.os.uk/downloads/open/BoundaryLine). **D** Warning scores of two random replicates with different cylinder volumes are strongly correlated, showing that outbreak detection is robust with respect to the cylinder volume choice. **E** Timeliness: updating a true-case warning score prospectively as new cases are added shows that it increases as the outbreak progresses, thus permitting detection earlier than the peak time t = 48 (colour shades are case dates as in A).

For comparison, we performed cluster detection with SaTScan [45] for retrospective space-time analysis using the discrete Poisson model [19]. SaTScan is a mature software widely used in disease surveillance that computes the scan statistics and is written in Java [45]. For our simulation experiment, the cylinder with the optimal likelihood ratio encloses 16 weeks (from 43 to 58) and 33 postcodes (Fig 6C, inset). These SaTScan results are widely consistent with ours; however, it is worth noting that SaTScan took around 3 hours to evaluate this sample dataset

on an Intel Core i5 CPU (excluding default Monte Carlo hypothesis testing and not accounting for seasonality, which would further increase the computational burden), thus significantly longer than the 5 minutes circa required by our R [46] implementation of RaNCover. As SaTS-can gives cylindrical boundaries of a detected cluster, these do not necessarily separate the outbreak cases from the baseline and, within these boundaries, there are cases correctly labelled only by RaNCover (Fig 6C).

## Application to iGAS disease: Cluster analysis and data-point embedding

Warning scores for all episodes of selected *emm* types are retrospectively generated following the procedure detailed in *Methods*, with $N = 20,000$ cylinders per *emm* type with respect to the baseline intensity of Eq (5). The spatial and temporal factors are obtained from census data and MLE fit, respectively. For each *emm* type considered, the type-specific factor $\lambda_\mathrm{m}$ is the relative frequencies of cases of that *emm* type (as illustrated in Fig 4 at each update time). As the average baseline intensity governs the cylinders' volumes, these are differentially set for each *emm* type, the rarer the *emm* type, the larger the volumes.

RaNCover consistently assigns high warning scores to neighbour cases and is thus able to reveal clusters which may belong to outbreaks. As each event is described by its spatial coordinates (e.g., latitude and longitude) and its date, the neighbourhood must be defined in both space and time. We use the t-distributed stochastic neighbour embedding (t-SNE) method to plot these cases (identified by three dimensions, i.e., latitude, longitude, and detection time) in a two-dimensional space in a way that preserves neighbouring identities [47]. In other words, t-SNE is a general statistical-learning method for dimensionality reduction which here is used to pin a two-dimensional point for each point in the three-dimensional space, in such a way that nearby events in space and time appear next to each other in the two-dimensional map. Using a visualisation method based on neighbourhoods rather than on absolute distances is very useful in our context, where the local prevalence of each *emm* type is important. Indeed, for *emm* types with low prevalence, recorded episodes that are distant in space and time can be correctly identified as being part of the same outbreak. Conversely, common *emm* types already established in the host population may show more clusters in the same geographical area and time scale. We illustrate the results for 4 rare *emm* types (94.0, 108.1, 44.0, and 33.0), and a common *emm* type that did not appear to cluster (12.0).

**emm type 94.0.** *emm* type 94.0 accounts for 466 invasive cases recorded over the 2015–2020 study period. Its frequency $\lambda_\mathrm{m}$ relative to all iGAS episodes stabilised in early 2018 after a phase of early growth (Fig 4B). Applying t-SNE to the episodes' spatial and temporal coordinates yielded the scatter plot of Fig 7A, which shows a cluster with episodes of high warning scores ($w > 0.95$) in bright red. These episodes are localised in both time (Fig 7B) and space (Fig 7C), and, specifically, may correspond to a small outbreak in the towns of Bournemouth and Weymouth (South West region) (Fig 7D) in 2017 (before $\lambda_\mathrm{m}$ reaches its plateau, see Fig 4B). Fig 7 also suggests the presence of other clusters of cases in the North West region, yet with $w < 0.95$, which might have triggered investigations but are not retrospectively tagged as of concern with a warning cut off at 0.95.

**emm type 108.1.** *emm* type 108.1, with 270 invasive cases recorded over the whole study period, was less common than the former. Yet this *emm* type is concerning as its prevalence has been steadily increasing since 2018 (Fig 4B) and in 2019 it accounted for more confirmed outbreaks than any other *emm* type [48]. For this *emm* type, RaNCover raised many warnings as the expected baseline is lowered by the absence of episodes prior to 2018 (Fig 8). The clusters with the highest warning scores include episodes in South East and South West, West Midlands, East Midland, and North West. Interestingly, in Yorkshire and the Humber region,

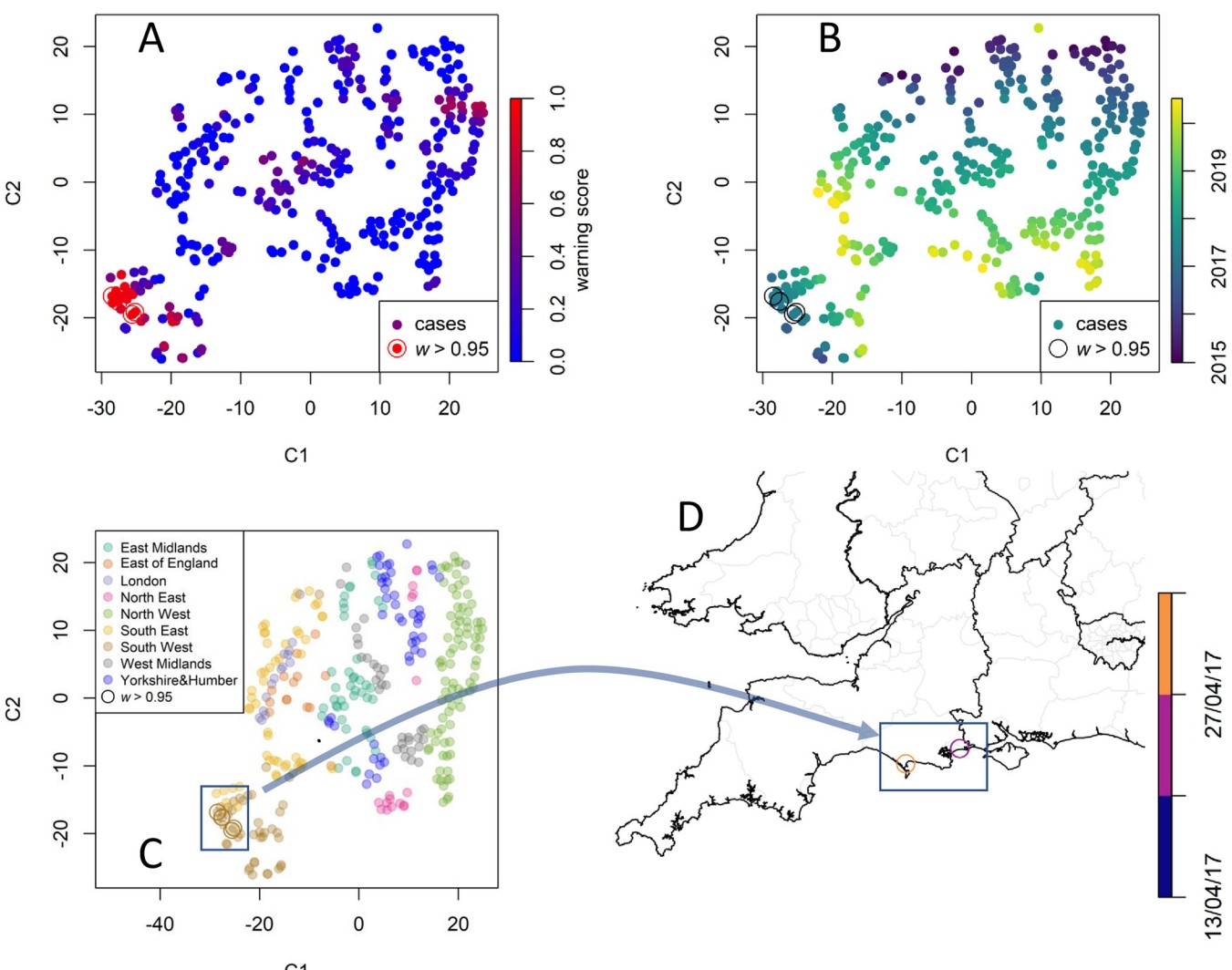

**Fig 7. Data point embedding and cluster analysis for emm type 94.0. A** Each iGAS record, identified by its position in time and space, is embedded in a two-dimensional space of coordinates $C_1$ and $C_2$ according to the t-SNE method. The points are coloured by their warning scores (0 to 1, blue to red), thus showing the presence of a bright red cluster of points with high warning scores ($w>0.95$). Colouring the detected cases by their record time (**B**) and region (**C**) shows that t-SNE preserves the neighbourhoods. **D** The framed cluster corresponds to cases localised in the towns of Bournemouth and Weymouth. Map created with Sf [55] using shapefiles from the GADM database (https://gadm.org/maps/GBR_1.html) and the Ordnance Survey Data Hub (https://osdatahub.os.uk/downloads/open/BoundaryLine).

we detect the presence of two clusters both localised in the city of Leeds but separated in time. Another cluster localised around the river Humber was also detected.

 **emm type 44.0.** *emm* type 44.0 totals only 163 recorded cases and its relative prevalence is nearly constant in time, except for a small increase in mid-2019 (Fig 4B). Consistently, RaN-Cover only detected a single cluster of episodes between March and August 2019 localised around the town of Chelmsford in the East of England (Fig 9). The cases in the South East and London are geographically distant but occurred at the same time as the bulk of the main cluster and are therefore mapped nearby the main cluster in Fig 9C.

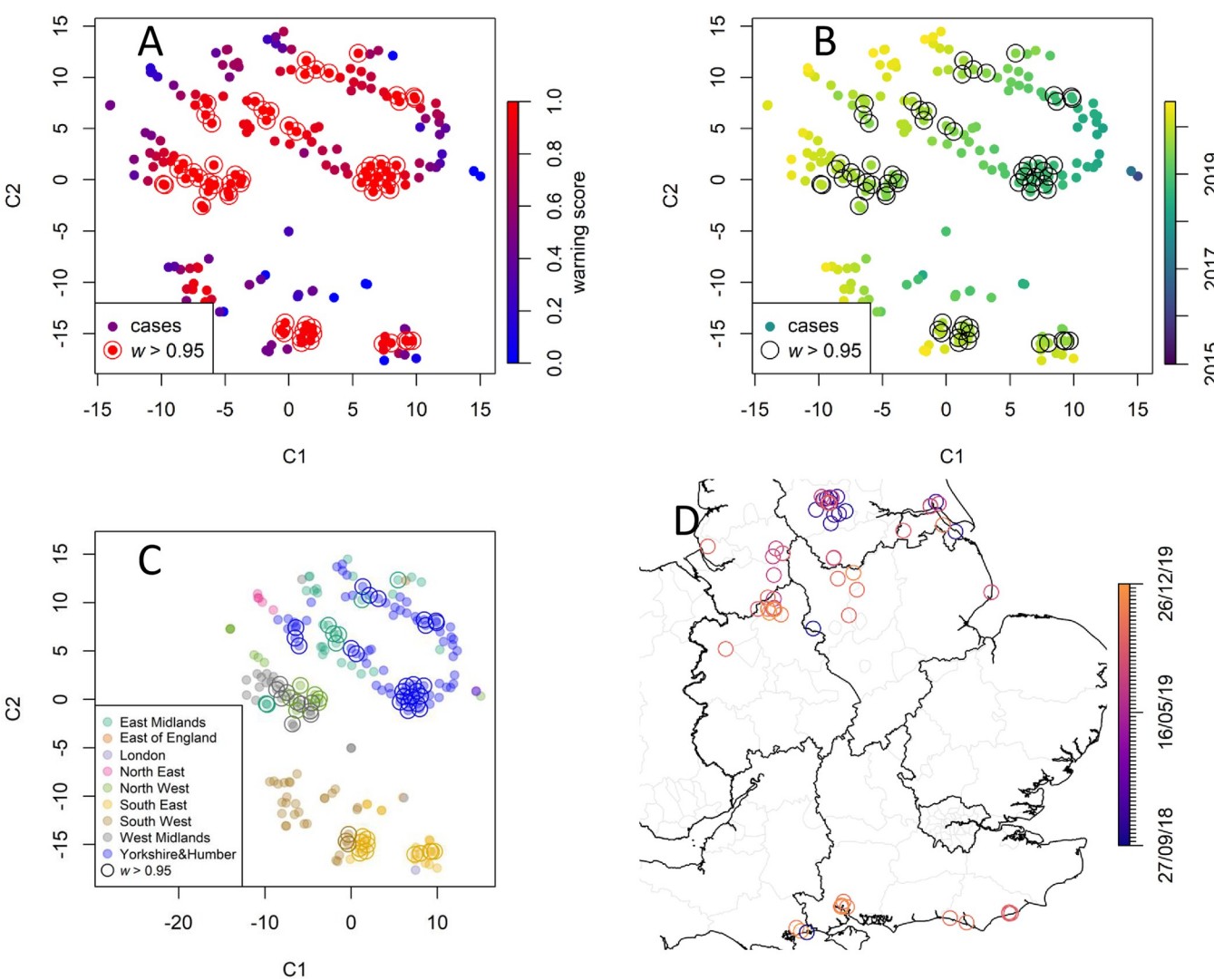

**Fig 8. Data point embedding and cluster analysis for emm type 108.0. Keys as in Fig 7.** emm 108.0 is a concerning type due to its recent increase in the number of cases. The many cases tagged with high warning scores (**A**) occurred after 2018 (**B**) and are clustered around several geographical locations (**C-D**). Map created with Sf [55] using shapefiles from the GADM database (https://gadm.org/maps/GBR_1.html) and the Ordnance Survey Data Hub (https://osdatahub.os.uk/downloads/open/BoundaryLine).

*emm* **type 33.0.** With a total of only 61 invasive cases recorded, the *emm* type 33.0 is uncommon in England. Its relative prevalence has been linearly growing since 2019 (Fig 4B). Similar to 108.1, RaNCover yielded high warning score for almost all detected cases. t-SNE embedding shows that these belong to a single cluster which corresponds to cases diffused in central England with epicenter in the West Midlands (Fig 10).

*emm* **type 12.0.** At the other end of the spectrum, the *emm* type 12.0 is relatively common (782 cases over the study period) and its trend has been decreasing since mid-2016. The application of RaNCover highlighted very few points which might raise concern with the majority of the episodes being assigned low warning scores and none being assigned a warning score higher than 0.95 (Fig 11).

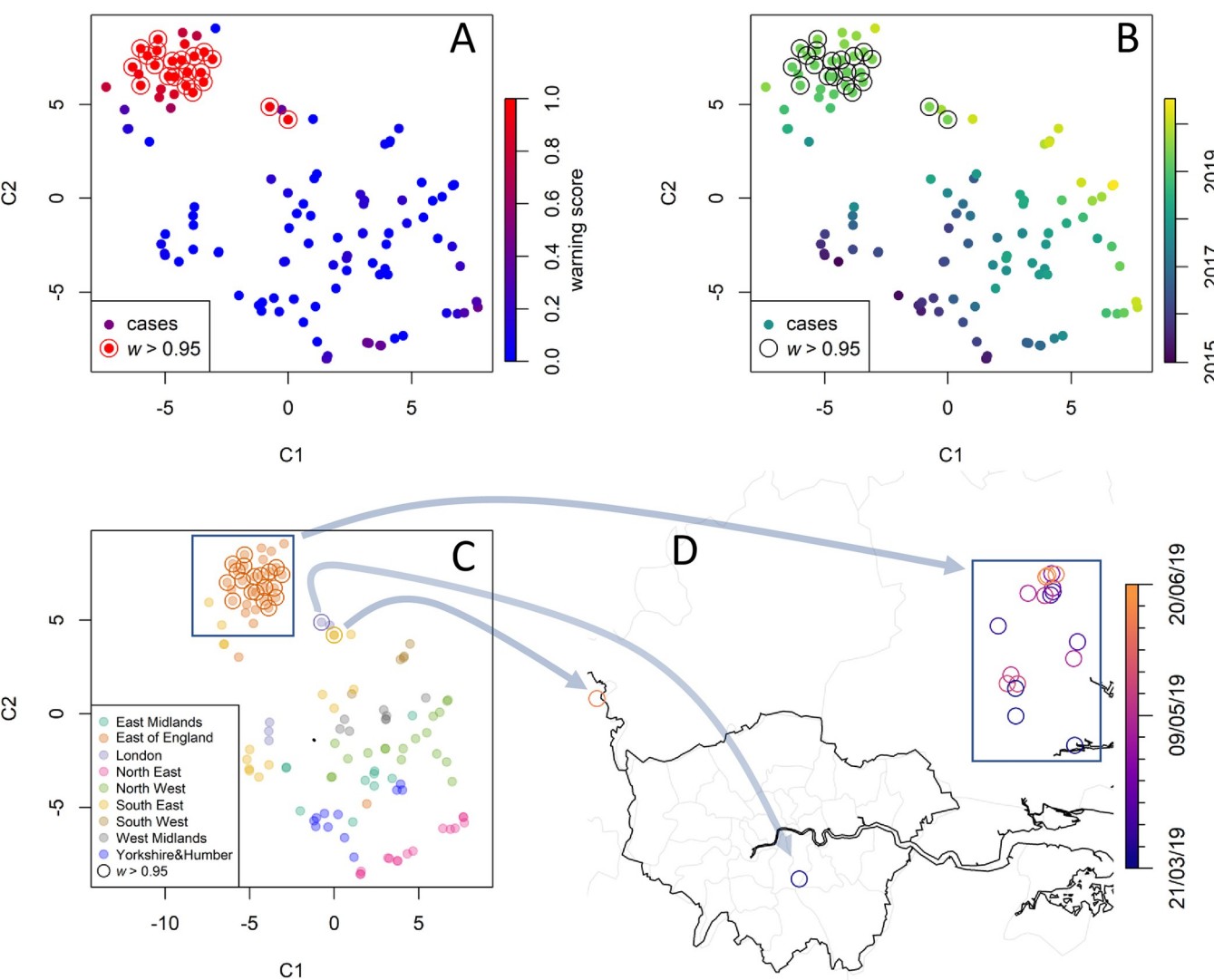

**Fig 9. Data point embedding and cluster analysis for emm type 44.0. Keys as in Fig 7.** The analysis detects a compact cluster localised in the East of England in 2019 (rectangle), with its embedded points mapped to their geographical location by the arrows. Map created with Sf [55] using shapefiles from the GADM database (https://gadm.org/maps/GBR_1.html) and the Ordnance Survey Data Hub (https://osdatahub.os.uk/downloads/open/BoundaryLine).

## Application to iGAS disease: Prospective analysis and *emm* typing delay

The cluster analysis detailed in the previous subsection is performed retrospectively and is based on cases labelled by the week at which the reference laboratory identified the pathogen's *emm* type. As anticipated for synthetic data (Figs 3 and 6E), it is also possible to observe single-case warning scores varying in time as new cases are *emm* typed and the outbreak progresses. As an example, we focus on the *emm* type 44.0, which appeared to have caused an outbreak in the East of England (Fig 9) with 23 cases retrospectively flagged at the end of the study period (circled red markers in Fig 9A and 9B). The warning scores computed at each week are illustrated in Fig 12A. As the first case of the cluster is detected, its score is high yet substantially below the threshold of 0.95. As more cases are detected, it approaches the threshold and

## Emm type 33.0

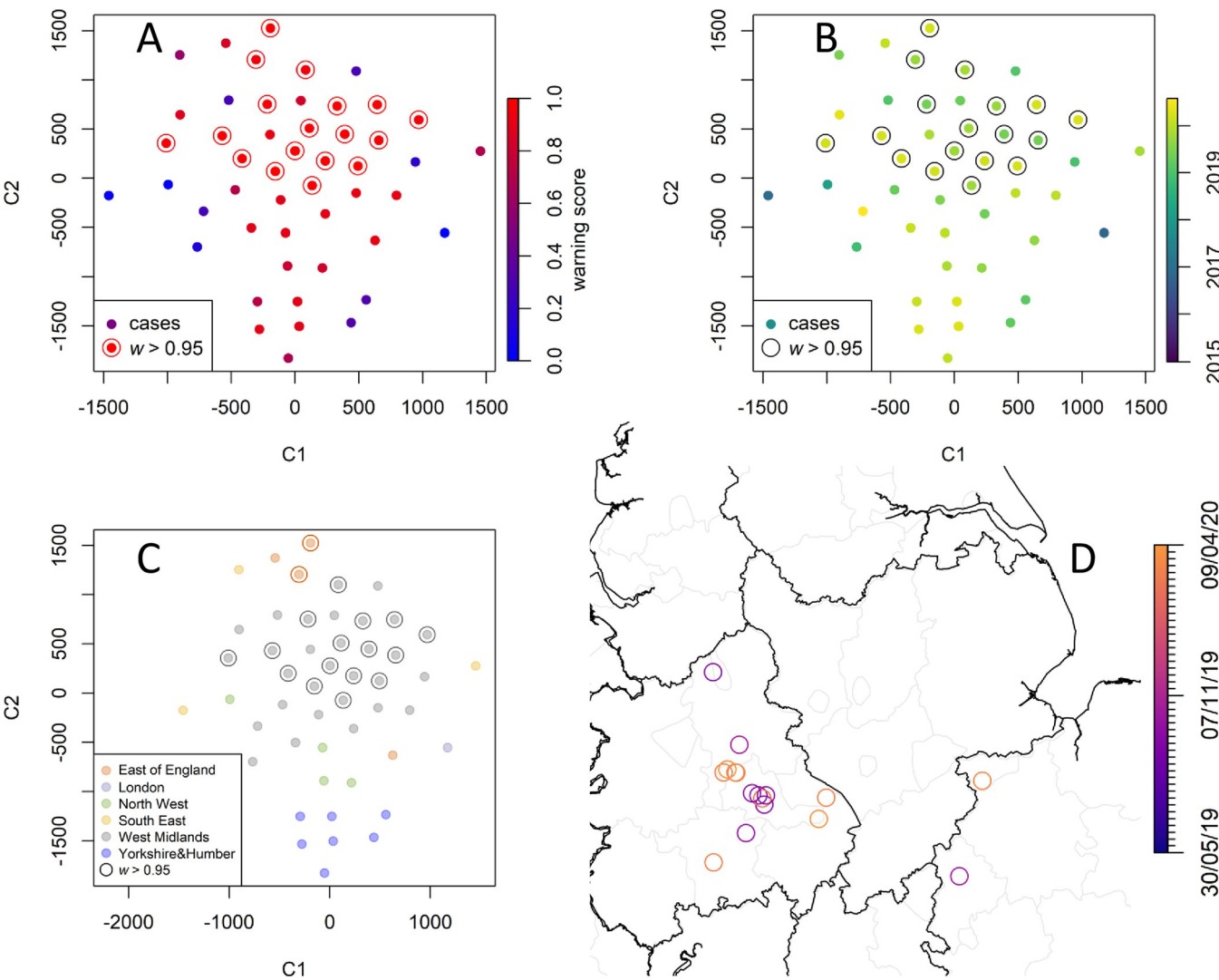

**Fig 10. Data point embedding and cluster analysis for the rare emm type 33.0. Keys as in Fig 7.** Inspection suggests that these cases belong to a single diffused cluster that emerged in September 2019. Map created with Sf [55] using shapefiles from the GADM database (https://gadm.org/maps/GBR_1.html) and the Ordnance Survey Data Hub (https://osdatahub.os.uk/downloads/open/BoundaryLine).

sharply increases within the first four weeks (Fig 12B), during which nine cases are identified in the region.

It is possible to combine reporting delays in RaNCover. During the interval from the sampling to the *emm* typing date, we are aware that, at a precise location and time, a case of an *emm* type of concern *might* have occurred. Therefore, only incomplete information is available. Even so, it is desirable to account for this, since it may allow earlier outbreak detection. However, it is possible that an untyped case is not actually associated with the clusters of interest, thus including it into a surveillance system may substantially tamper the overall results. It is important to choose a suitable strategy, which we take to flag a subset if the untyped or typed observation exceeds their respective baselines. Prospectively, as the time progresses, new cases are observed and some previously observed cases are *emm* typed. In such a scenario, at a given

## Emm type 12.0

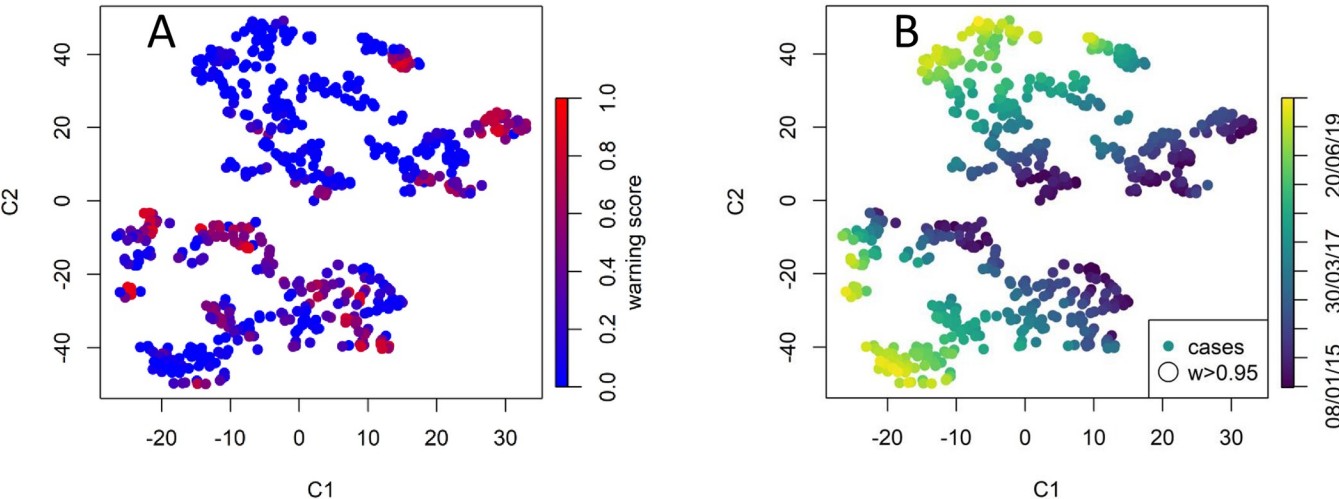

**Fig 11. Data point embedding for the common emm type 12.0.** No detected case is assigned a warning score higher than 0.95.

point in time, in addition to the typed iGAS cases and their baselines (one for each *emm* type) one can collect the untyped cases and compute the corresponding baseline. As an example, the number of cases per week and the continuous baselines are plotted against time in Fig 12C for the *emm* type 44.0—similar snapshots can be obtained for any other point in time. The number of untyped cases is highest in correspondence of the update week and quickly decays to zero as the observation week goes back in time. Using these two baselines and according to the procedure detailed in *Methods*, we computed the warning scores relative to a potential *emm* 44.0 outbreak of the untyped cases at each update week. More specifically, we are interested in the early warning scores of those cases which are later found to belong to the 2019 cluster. The early warning scores did not reach the threshold of 0.95, yet they are significantly larger than the baseline warning scores thus potentially providing early warning signals as illustrated in Fig 12D.

## Discussion

In this paper, we have presented the random neighbourhood covering (RaNCover), a statistical method that can identify outbreaks by detecting deviations from a baseline Poisson point process. It is based on a simple and scalable approach, i.e., on measuring the expected and observed numbers of cases in random portions (such as cylinders) of space and time. While the statistics in a single random region are not highly informative, aggregating information over many random regions permits the detection of infectious-disease clusters. A strength of this approach is that it returns a warning score for each recorded episode. This makes it ideal in application to diseases which are sparsely distributed, where it is appropriate to assess a new health event as soon as it is recorded, rather than waiting until enough cases are collected over a determined region or time interval.

Our method is akin to those based on the so-called scan statistics [16]. They both rely on creating a large number of spatio-temporal sectors, each with a different set of detected cases in it. By means of the scan statistics one can identify the sectors with the highest likelihood ratio, which correspond to the regions where outbreaks are most likely to have occurred.

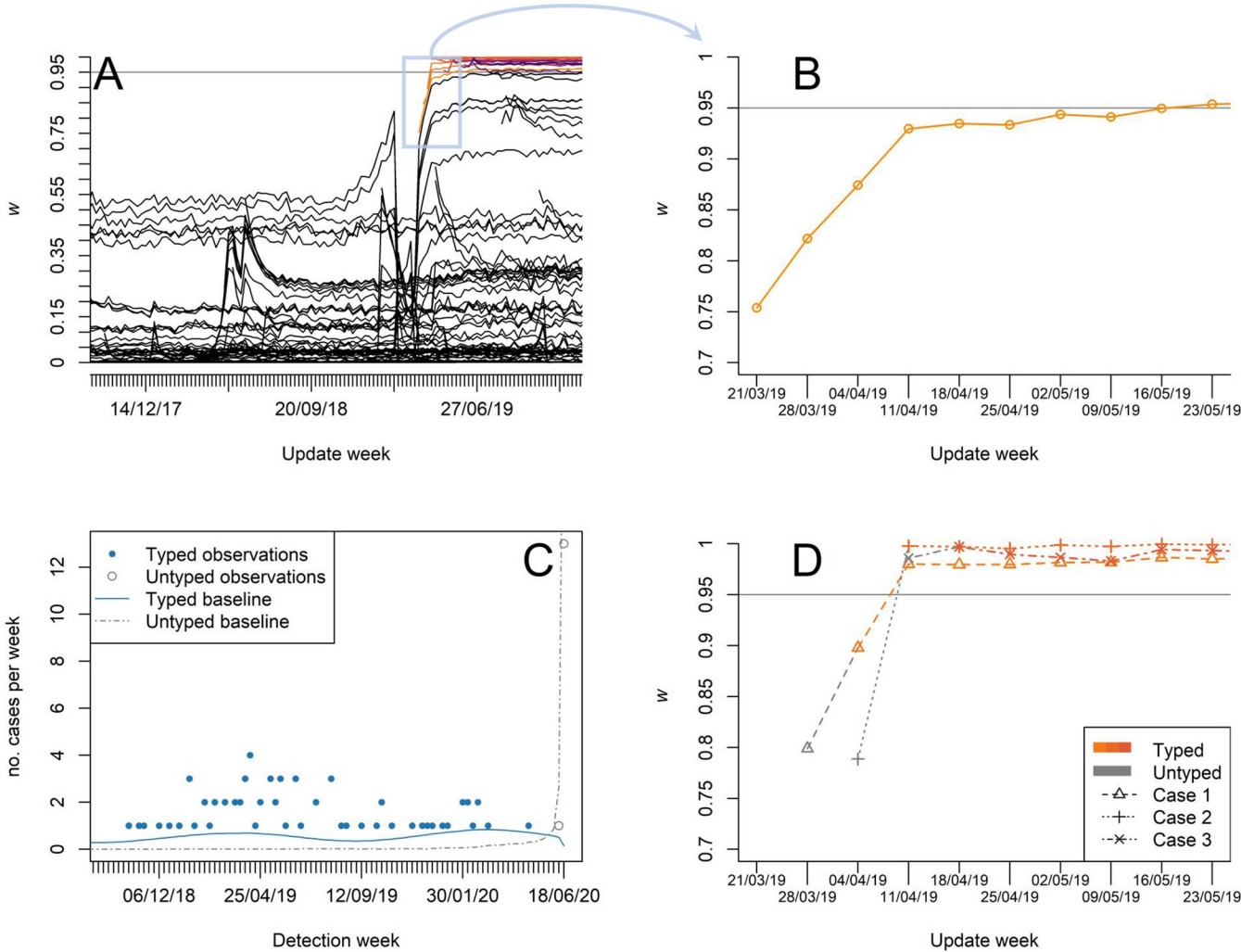

**Fig 12. Prospective analysis for emm type 44.0. A** Warning scores retrospectively computed every week as new cases are included; some cases, corresponding to the cluster of March 2019 in the East of England, incur a sharp increase in their warning scores (top right lines, amber to violet colour indicate typing week). **B** The score of the first cluster case (detected and typed in the week of 21/03/19) increases rapidly during the first four weeks. **C** Observation counts (marker) and estimated baselines (lines) in the week of 18/06/20 for both typed (blue colour) and untyped (grey colour) cases. **D** Warning scores of selected cluster cases. These cases were initially recorded without any typing information. Using the untyped observations, it is possible to obtain early warning scores (grey colour). The samples are typed a week after detection and the warning scores are updated accordingly (amber to orange colour shade indicate type week).

Conversely, in RaNCover, drawing and inspecting many cylinders covering each event **x** permits defining a continuous warning score *w* between 0 and 1 specific to **x**. This permits classification of cases in two groups, one for outbreak and one for baseline, with *w* corresponding to the probability that a case is outbreak-related. Compared to existing methods, it excels in execution speed and flexibility. One potential limitation of our approach is that the values of the warning scores depend on the chosen volumes of the spatio-temporal cylinders and are subject to fluctuations due to random sampling. In fact, extensive simulations show that, performing replicates with different cylinder sizes, the results remain consistent and fluctuations can be controlled by increasing the sampling.

RaNCover can be used for both retrospective and prospective analyses. Retrospective validation was performed on simulated data using Receiver Operating Characteristic (ROC) curves. Deriving a binary warning from *w* requires choosing a cut off, which also governs

 

sensitivity and specificity. For a given cut off, the sensitivity is defined here as the fraction of simulation-outbreak cases for which an alarm is generated, while the specificity is defined as the fraction of baseline cases that did not generate an alarm. The overall performances quantified by the ROC curves (obtained by plotting the sensitivity against the specificity at various cut-off settings) and the area under the curves (AUCs or C-statistics) were excellent (ROC-AUCs >0.9 in all simulation scenarios). Fine tuning the cut off also serves as a way of reducing the number of false alarms or increasing the true positive rate, which can be useful in real-world surveillance settings.

As a proof-of-concept, we retrospectively detected a number of potential outbreaks of iGAS infections in England. For this disease, we estimated the baseline taking into account population denominator data and iGAS seasonality. In the absence of a definitive and comprehensive list of all real outbreaks that should be detected by public-health surveillance, it is hard to fully benchmark any surveillance method. In order to circumvent this latter point and confirm that our method consistently detects events in close proximity (both in space and time), we proposed to visualise the data using t-distributed stochastic neighbour embedding (t-SNE) approach. The use of this dimensionality reduction algorithm on real iGAS data highlights events proximal in both space and time and confirms that detected cases with high warning scores typically cluster together.

The timeliness was prospectively assessed by tracking how the individual warning scores vary as new observations are included. It is worth noting that, in general, the timeliness can be influenced by the speed of the outbreak growth [38]. It can also be influenced by the outbreak type and its full assessment requires comprehensive testing with different outbreak scenarios, as performed for established methods, e.g., in references [38,43,49,50]. Reporting delays are also known to affect surveillance practice. These can differ substantially between pathogens (which may require laboratory tests of different complexity) and between laboratories; some identifying some specimens may need several steps in succession; reporting delays may even vary over time, thus making relevant to address these with a general approach [51–53]. It is desirable to leverage the information on an infection case potentially caused by the pathogen of interest, as this can anticipate the complete detection of a cluster upon *emm* typing. As an approach to this, we thus defined an early warning score based on exceedances in both typed and untyped records in the neighbourhoods of a case of interest. For cases that were later identified as part of a cluster, these early scores were indeed higher than those in the background, thus suggesting good predictive power.

Further work is planned to validate RaNCover in public health practice. This is based on the compilation of iGAS outbreak incidents identified through a variety of consensus sources, including whole-genome sequencing, contact tracing, and detailed public health investigation, which can ultimately confirm the presence of epidemiological links. It will be also interesting to explore and test variants of this methodology, similarly to how several variants of the scan statistics have been proposed (and implemented in statistical software) since the scan statistics was first introduced in 1965 [17,18].

In conclusion, the RaNCover method is a convenient clustering tool for the spatio-temporal surveillance of communicable diseases such as iGAS infections, which typically are sparsely distributed over a geographical area and yet pose major threats to people's health due to their severity. For these diseases, traditional surveillance methods typically require aggregating data over time and space. Consequently, they cannot fully leverage all the detailed geographical information that is available. Similarly to SaTScan, our method can be used at arbitrary levels of geographical and temporal aggregation. RaNCover further provides a continuous warning score for each detected case, rather than for coarse-grained time windows or geographical regions, and yet remains able to highlight emerging clusters of diseases. The method only

 

requires deriving a null model for the baseline and can be used for different types of diseases, regardless of the complexity of their host-pathogen dynamics.

## Supporting information

**S1 Fig. Retrospective assessment of RaNCover performances by means of simulations. A-B** ROC curves (solid black line) of RaNCover predictions over two replicates, obtained using different cylinder volumes. Solid grey curves are from ROCs obtained by means of bootstrapping. The dotted red lines intercept the values of sensitivity (horizontal lines, sensitivities equal to 0.54 and 0.40 for **A** and **B**, respectively) and specificity (vertical lines, specificity equal to 0.94 and 0.98 for **A** and **B**, respectively) obtained from setting $1-\alpha = 0.95$ as a discrimination threshold. **C** Correlation plot between warning scores from the two replicates. Red markers correspond to true epidemic simulation events.
(TIF)

**S2 Fig. Illustration of simulation-experiment data aggregated and projected to time (A-B) and geography (C-D).** Cases are simulated from a Poisson point process with endemic intensity function fitted on true iGAS data, thus encoding for seasonality (**A**) and population density (**C**). Other cases representing a small outbreak occurring from t = 40 to t = 59 in St Albans (Hertfordshire, highlighted areas in **A,B,C**, and **D**) are included in the synthetic dataset; the overall temporal and geographical patterns are not affected (**B** and **D**), yet the algorithm is able to spot the anomaly (Fig 6, main text). Maps created with Sf [53] using shapefiles from the GADM database (www.gadm.org) and the Ordnance Survey Data Hub (osdatahub.os.uk). Same data embedded in a two-dimensional plane using t-SNE are illustrated in S4 Fig.
(TIF)

**S3 Fig. Confidence intervals of warning scores.** Widths of confidence intervals (CIs) of warning scores in simulation experiment for two choices of total number of random cylinders drew ($N = 10^6$ and $N = 5 \times 10^5$, solid and dashed lines, respectively) and two choices of cylinders' volume (V1 and V2 = V1×1.4, blue and grey lines, respectively). The warning scores are proportions of flagged cylinders and appropriate CIs are Wilson's for binomial proportions. Increasing N or increasing the volumes, the numbers at numerator and denominator of the proportion increase and the confidence in its estimate improves.
(TIF)

**S4 Fig. Illustration of simulation data projected on a two-dimensional plane using t-distributed stochastic neighbour embedding (t-SNE).** Each simulated record is identified by a point of coordinates C1 and C2. The points are coloured by their warning scores ((**A**) 0 to 1, blue to red, highlighting the presence of a bright red cluster of points with high warning scores $w > 0.95$) and by their record time (**B**), showing that t-SNE also preserves temporal proximity.
(TIF)

## Acknowledgments

The authors would like to thank colleagues in microbiology laboratories for their prompt reporting of iGAS cases and submission of isolates, the UKHSA streptococcal reference laboratory (Karen Broughton, Roger Daniel, and Chenchal Dhami), and the wider collaborative group working on iGAS cluster detection, which incorporates Consultants in Communicable Disease, Field Epidemiology Consultants, Topic Experts, and Public Health Registrars. We also thank Majel McGranaham and Aoife Barror for valuable discussions.

## Software and reproducibility

Analyses were carried out using R ([46], version 3.4.3). t-distributed stochastic neighbour embedding was performed using the package Rtsne [54] with default parameters. Maps created with Sf [55] using shapefiles from the GADM database (www.gadm.org) and the Ordnance Survey Data Hub (osdatahub.os.uk). All software (also including codes for generating of synthetic data) is archived online at https://github.com/mcavallaro/outbreak-detection.

## Author Contributions

**Conceptualization:** Massimo Cavallaro, Noel D. McCarthy, Dan Todkill, Matt J. Keeling.

**Data curation:** Massimo Cavallaro, Juliana Coelho, Derren Ready, Valerie Decraene, Theresa Lamagni, Dan Todkill.

**Formal analysis:** Massimo Cavallaro, Matt J. Keeling.

**Funding acquisition:** Noel D. McCarthy, Matt J. Keeling.

**Investigation:** Massimo Cavallaro.

**Methodology:** Matt J. Keeling.

**Software:** Massimo Cavallaro.

**Validation:** Massimo Cavallaro.

**Visualization:** Massimo Cavallaro.

**Writing – original draft:** Massimo Cavallaro, Matt J. Keeling.

**Writing – review & editing:** Massimo Cavallaro, Juliana Coelho, Derren Ready, Valerie Decraene, Theresa Lamagni, Noel D. McCarthy, Dan Todkill, Matt J. Keeling.

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
