## [Decision Letter · Decision Letter 0]

21 Mar 2022

Dear Dr. Cavallaro,

Thank you very much for submitting your manuscript "Cluster detection with random neighbourhood covering: application to invasive Group A Streptococcal disease." for consideration at PLOS Computational Biology.

As with all papers reviewed by the journal, your manuscript was reviewed by members of the editorial board and by several independent reviewers. In light of the reviews (below this email), we would like to invite the resubmission of a significantly-revised version that takes into account the reviewers' comments.

We cannot make any decision about publication until we have seen the revised manuscript and your response to the reviewers' comments. Your revised manuscript is also likely to be sent to reviewers for further evaluation.

Sincerely,

Roger Dimitri Kouyos

Associate Editor

PLOS Computational Biology

Nina Fefferman

Deputy Editor

PLOS Computational Biology

Reviewer's Responses to Questions

**Comments to the Authors:**

Reviewer #1: The authors introduce a new method (RaNCover) intended, after further validation, to be applied for spatiotemporal surveillance of communicable diseases. I review from the perspective of a public health practitioner and epidemiologist with experience applying the scan statistics in SaTScan for prospective outbreak detection. I advise this paper also be evaluated by a reviewer with more biostatistical expertise than I.

1. P. 7: “The general problem we address here is to assess whether a single recorded infection might be part of a cluster or is consistent with a baseline of sporadic cases:” this seems like a non sequitur. Could the authors please clarify the logical flow from the rest of the Introduction to the statement of the practical problem this new method was created to address? In jurisdictions where disease counts are so sparse that RaNCover should be run after every new case is reported, then there can be eyes on every case to assess whether an outbreak is unfolding. It seems that running prospective analyses with data aggregated to a daily time interval should be sufficient for timely iGAS outbreak detection.

2. P. 9: “standard false discovery rate = 0.05:” This is standard for retrospective analyses, but for repeated, prospective analyses as the authors propose, this will cause too much false signaling. Recurrence intervals > 1 year are more standard thresholds. (See, e.g., SaTScan user manual, “recurrence interval.”)

3. P. 15, “Prospective analysis and emm typing delay” section: Do the authors propose health departments implement one prospective analysis for all iGAS data regardless of typing, a separate prospective analysis for every emm type, and/or a separate prospective analysis for every emm type and all untyped cases? For operational simplicity, is it possible to run one daily analysis for iGAS, accounting for both emm type (if available) and untyped cases?

4. Could the authors please clarify if and how iGAS seasonality patterns vary across emm types?

5. Could the authors provide recommendations for choosing the length of baseline period?

6. Is it correct that iGAS disease data were obtained for 2015-2020, but the population denominators were from 2011 (P. 11)? If so, why did the authors use outdated estimated population sizes?

7. Could the authors please comment on strengths and limitations for using RaNCover with population denominator data (which can quickly become outdated for small areas), when there are alternative methods readily available (e.g., space-time permutation scan statistic, Ref #19) that use disease counts in the baseline period to establish expected case counts, with no requirement for population denominators?

8. In the statements on ethics and software and reproducibility, I suggest the authors clarify the study also used synthetic data, as posted to their GitHub site.

9. Fig. 5: could the authors please comment re: whether the large spike in 2018 above baseline was a true outbreak? What is the explanation for this increase?

10. P. 13: re: the SaTScan analysis “not accounting for seasonality, which would further increase the computational burden:” Please clarify why the authors chose to apply the discrete Poisson model. The space-time permutation model (Ref #19) would have accounted for purely spatial and purely temporal clusters such as seasonal effects.

11. P. 16, “Compared to existing methods, [RaNCover] excels in execution speed and flexibility.” First, 3 hours is not a prohibitively long run-time, since programs can be automated and run overnight to be complete by open of business. Second, could the authors please clarify how RaNCover is more flexible than the scan statistics, and how this flexibility adds practical value?

12. P. 17: “traditional surveillance methods typically require aggregating data over time and space:” In our applications of scan statistics, we aggregate to daily temporal resolution and (depending on the application) either census tract spatial resolution or the exact X-Y coordinates of the patient address (i.e., no spatial aggregation necessary). Please edit to avoid suggesting that other methods require use of “coarse-grained time windows or geographical regions.” Further, it appears the authors aggregated to “patient postcode location” (P. 11), so it appears that unlike some of the scan statistics, RaNCover has the limitation of requiring aggregating to geographical regions with accurate population denominator data.

13. Could the authors please explain more explicitly why the method is named “RaNCover,” i.e., in what sense are neighborhoods being covered randomly?

Reviewer #2: The study entitled “Cluster detection with random neighbourhood covering: application to invasive Group A Streptococcal disease” by Cavallaro et al. presents a very interesting approach to detect epidemic events based on spatio-temporal data. It demonstrates how the method can be applied to detect outbreaks of invasive GAS, integrating emm-typing information. The latter point is of special importance, as such level of detail has been previously ignored in several epidemiological models. Approaching the complexity of the real picture as much as possible, including delay of typing, is a merit of this study. The manuscript is clear and well written, and code is available.

Additional comments:

- Although the iGAS cases across England are confidential data, it is very helpful to be able to run the code of simulation_experiment_England.Rmd.

Unfortunately, the file including geographical location data (shape_file_new/Areas.shp) is not available… A comment implies that sharing of the UK boundaries shape files might not be allowed… Is it not?

- The dimensionality reduction method ‘t-distributed stochastic neighborhood embedding’ is used in most figures of the paper. Please give more in details regarding the algorithm used, the parameters considered (is it only the three variables x, y, time, or other variables?).

- Linking the two previous points, would it be possible to add two panels to Fig. 6 displaying the simulation data as they are displayed in the A-B panels of the following figures? (it would be slightly redundant with the current panels B and C, so doing so in an additional supplementary figure might be the best)

- Equation 3) there is a parameter E, that is not mentioned in the following definition

- Please define timeliness

- For four rare emm-types, outbreak clusters were identified, while for the only common emm-type examined, no outbreak cluster was identified. Is this a coincidence? Is there a specific reason why the other two common emm-type (emm-1.0 and emm-89.0) were not examined? Can the authors elaborate on detection efficiency depending on lambdam? and/or on the implication of an increasing lambdam in the years considered?

- “As anticipated in synthetic data...” the authors probably refer to figure 3 and 6E, and similarly in the legend of Fig S2, “the algorithm is able to spot the anomaly …” the authors probably refer to Fig 6

- Fig. 12 the x-axes range and labels change in each panel make the figure difficult to read. Having same range/labels in B/D and highlight which portion of A/C they correspond to may facilitate understanding.

- Fig S2 makes an important point very clear, but adding a visual guide pointing towards the actual differences (e.g shading of the timeframe 40-60 hours and the region of St. Albans) would be helpful.

Reviewer #3: The topic addressed in this article is extremely important for epidemiological surveillance in general and for the detection of infectious disease clusters in particular. The introduction is very clear but the article suffers from three major shortcomings concerning the description of the method, the simulation study and the scripts.

The method would need to be structured in sub-sections and probably lacks a part to help users make choices on tuning parameters. For example, how to choose a value for N in practice? What does "on average greater than one" or "contains on average more than one basic event" mean? The choice of the volume constant is not clear. What is the impact of the choice of tau on the performance of the method?

The simulation study is not well designed. The proposed method needs to be evaluated by a thorough simulation study, which could be described in detail in an appendix. In this respect, the authors can follow the recommendations proposed in this article: Using simulation studies to evaluate statistical methods (Morris, White, Crowther) Statistics in Medicine 2017. A thorough simulation study is the only way to scientifically validate the proposed method and this is not the case in the manuscript.

The scripts that are available on Github do not allow to run the method. Some scripts (source programs) are missing and some data are also missing. It is thus not possible to test its behaviour using other real data. It would be very helpful if the authors would provide a main program that works with simulated data.

**Have the authors made all data and (if applicable) computational code underlying the findings in their manuscript fully available?**

Reviewer #1: Yes

Reviewer #2: Yes

Reviewer #3: **No: **The scripts that are available on Github do not allow to run the method.

PLOS authors have the option to publish the peer review history of their article (what does this mean?). If published, this will include your full peer review and any attached files.

Reviewer #1: No

Reviewer #2: No

Reviewer #3: No
---

## [Decision Letter · Decision Letter 1]

15 Sep 2022

Dear Dr. Cavallaro,

Thank you very much for submitting your manuscript "Cluster detection with random neighbourhood covering: application to invasive Group A Streptococcal disease." for consideration at PLOS Computational Biology. As with all papers reviewed by the journal, your manuscript was reviewed by members of the editorial board and by several independent reviewers. The reviewers appreciated the attention to an important topic. Based on the reviews, we are likely to accept this manuscript for publication, providing that you modify the manuscript according to the review recommendations.

Sincerely,

Roger Dimitri Kouyos

Associate Editor

PLOS Computational Biology

Nina Fefferman

Deputy Editor

PLOS Computational Biology

Jason A. Papin

Editor-in-Chief

PLOS Computational Biology

Feilim Mac Gabhann

Editor-in-Chief

PLOS Computational Biology

[LINK]

Reviewer's Responses to Questions

**Comments to the Authors:**

Reviewer #1: I thank the authors for carefully considering my first round of comments. I reiterate that I review from the perspective of an applied public health practitioner and am not a statistician so I am perhaps not your primary audience.

I would like to comment on the responses to Reviewer #1 based on my experience with SaTScan because the contrasts with SaTScan in this manuscript can support public health practitioners in choosing which methods to apply in practice:

#7, re: “We think that methods for outbreak detection should take spatial population density data into account or otherwise suffer from heavy biases towards locations with high population density, such as metropolitan areas, although we are aware that this may not work with any surveillance data.”

- In the absence of large population movement during the study period, areas that have large case counts during the temporal window of interest because of high population density also have large case counts in the baseline period, so the expected number of cases for any area implicitly accounts for the population distribution, avoiding the “heavy biases” the authors mention. This has been my experience in applying the space-time permutation scan statistic.

- This relates to my main remaining question after reading through the revision: could the authors please provide more intuition about how RaNCover is able to classify every single recorded infection as cluster-related vs. sporadic for areas with high population density where the baseline level of endemic disease is high?

#10, re: “Also, it appears to us that SaTScan with discrete Poisson model can in principle account for seasonality but with substantially increased computational burden. We believe this computational burden is intrinsic to SaTScan and we expect that SaTScan with space-time permutation model would suffer from the same issue.”

- Per the SaTScan user manual: “The space-time permutation model automatically adjusts for purely spatial and purely temporal clusters [seasonal trends are purely temporal clusters]. For the discrete Poisson model, purely temporal and purely spatial clusters can be adjusted for in a number of different ways [options for temporal trend adjustments, e.g., for seasonality, include nonparametric, log-linear, or log-quadratic].” I believe the computational burden with the discrete Poisson model depends on the choice of temporal trend adjustment but in my experience has not resulted in prohibitively increased computational burden. The space-time permutation model does not suffer from increased computational burden with the addition of seasonal adjustments because adjustment for purely temporal trends including seasonality is automatic/implicit in the probability model.

#11, re: “methods based on the scan statistics outputs an outbreak area of a determined shape such as a cylinder.”

- Effective SaTScan v.10.0 (see SaTScan version history under technical documentation), users can detect clusters along a network (see “Network file” in user guide), not only determined shapes such as circles or ellipses.

Minor notes:

Intro: suggest clarifying common modes of transmission for iGAS outbreaks.

Lines 83-84: suggest avoiding the term “epidemiologically linked” in this sentence as seemingly refers to only spatiotemporal proximity. I agree with the different use of this term in line 524.

Line 91: suggest clarifying what is meant by “appropriately” as I cannot infer what an inappropriate random collection of cylinders might look like.

Line 257: How common is it for a single patient to be infected with multiple iGAS emm types?

Line 312, re: endemic cases “should not raise any public health concerns:” suggest rewording because high levels of preventable, endemic disease should be concerning to public health authorities, even in the absence of outbreaks.

Fig. 1: Is It intentional that all circles have the same radius? Line 196 refers to "variable radia" so I would have expected the illustration to have different radii (and isn't the plural of radius radii, not radia?).

Typos:

Line 25: “odd” should be “odds”

Line 291: “observation” should be “observations”

Line 815: “initial” should be “initially”

Line 816: “sample” should be “samples”

Reviewer #2: I thank the authors for their thorough revision. In particular, being able to run the code for the simulation experiment is very valuable. The methods are also much clearer.

A couple details:

-concerning your answer to my point for Figure 12, I agree that the highlight of the portion of A to which B corresponds is helpful, but I do not see a change in the x axes of panel B and D. I think shifting the plot of D to the right by one week, so that same dates are vertically aligned (in B and D) would make this plot more intuitive to read.

-I appreciate the addition of Figure S4, but don't forget to refer to it in the main text, linking it to Figure 6.

**Have the authors made all data and (if applicable) computational code underlying the findings in their manuscript fully available?**

Reviewer #1: Yes

Reviewer #2: Yes

PLOS authors have the option to publish the peer review history of their article (what does this mean?). If published, this will include your full peer review and any attached files.

Reviewer #1: No

Reviewer #2: No

Figure Files:

Data Requirements:

Reproducibility:

References:

---

## [Editor Report · Decision Letter 2]

10 Nov 2022

Dear Dr. Cavallaro,

We are pleased to inform you that your manuscript 'Cluster detection with random neighbourhood covering: application to invasive Group A Streptococcal disease.' has been provisionally accepted for publication in PLOS Computational Biology.

Best regards,

Roger Dimitri Kouyos

Academic Editor

PLOS Computational Biology

Nina Fefferman

Section Editor

PLOS Computational Biology

Jason A. Papin

Editor-in-Chief

PLOS Computational Biology

Feilim Mac Gabhann

Editor-in-Chief 

PLOS Computational Biology

---

## [Editor Report · Acceptance letter]

24 Nov 2022

PCOMPBIOL-D-21-01924R2 

Cluster detection with random neighbourhood covering: application to invasive Group A Streptococcal disease.

Dear Dr Cavallaro,

I am pleased to inform you that your manuscript has been formally accepted for publication in PLOS Computational Biology. Your manuscript is now with our production department and you will be notified of the publication date in due course.

With kind regards,

Orsolya Voros
